# Function and Fiber-Type Specific Distribution of Hsp60 and αB-Crystallin in Skeletal Muscles: Role of Physical Exercise

**DOI:** 10.3390/biology10020077

**Published:** 2021-01-21

**Authors:** Daniela D’Amico, Roberto Fiore, Daniela Caporossi, Valentina Di Felice, Francesco Cappello, Ivan Dimauro, Rosario Barone

**Affiliations:** 1Human Anatomy Section, Department of Biomedicine, Neuroscience and Advanced Diagnostics (BiND), University of Palermo, 90127 Palermo, Italy; damicoda90@gmail.com (D.D.); valentina.difelice@unipa.it (V.D.F.); 2Department of Neuroscience, Cell Biology, and Anatomy, University of Texas Medical Branch (UTMB), Galveston, TX 77554, USA; 3Postgraduate School of Sports Medicine, University Hospital of Palermo, 90127 Palermo, Italy; roberto.fiore@mail.com; 4Department of Movement, Human and Health Sciences, University of Rome Foro Italico, 00135 Rome, Italy; daniela.caporossi@uniroma4.it; 5Euro-Mediterranean Institutes of Science and Technology (IEMEST), 90139 Palermo, Italy

**Keywords:** skeletal muscle, physical exercise, heat shock protein 60, CRYAB, myosin heavy chain

## Abstract

**Simple Summary:**

Skeletal muscle represents about 40% of the body mass in humans and it is a copious and plastic tissue, rich in proteins that are subject to continuous rearrangements. Physical exercise is considered a physiological stressor for different organs, in particular for skeletal muscle, and it is a factor able to stimulate the cellular remodeling processes related to the phenomenon of adaptation. All cells respond to various stress conditions by up-regulating the expression and/or activation of a group of proteins called heat shock proteins (HSPs). Although their expression is induced by several stimuli, they are commonly recognized as HSPs due to the first experiments showing their increased transcription after application of heat shock. These proteins are molecular chaperones mainly involved in assisting protein transport and folding, assembling multimolecular complexes, and triggering protein degradation by proteasome. Among the HSPs, a special attention needs to be devoted to Hsp60 and αB-crystallin, proteins constitutively expressed in the skeletal muscle, where they are known to be important in muscle physiopathology. Therefore, here we provide a critical update on their role in skeletal muscle fibers after physical exercise, highlighting the control of their expression, their biological function, and their specific distribution within skeletal muscle fiber-types.

**Abstract:**

Skeletal muscle is a plastic and complex tissue, rich in proteins that are subject to continuous rearrangements. Skeletal muscle homeostasis can be affected by different types of stresses, including physical activity, a physiological stressor able to stimulate a robust increase in different heat shock proteins (HSPs). The modulation of these proteins appears to be fundamental in facilitating the cellular remodeling processes related to the phenomenon of training adaptations such as hypertrophy, increased oxidative capacity, and mitochondrial activity. Among the HSPs, a special attention needs to be devoted to Hsp60 and αB-crystallin (CRYAB), proteins constitutively expressed in the skeletal muscle, where their specific features could be highly relevant in understanding the impact of different volumes of training regimes on myofiber types and in explaining the complex picture of exercise-induced mechanical strain and damaging conditions on fiber population. This knowledge could lead to a better personalization of training protocols with an optimal non-harmful workload in populations of individuals with different needs and healthy status. Here, we introduce for the first time to the reader these peculiar HSPs from the perspective of exercise response, highlighting the control of their expression, biological function, and specific distribution within skeletal muscle fiber-types.

## 1. Introduction

All organisms and cells respond to various stress conditions by up-regulating the expression and/or activation of a group of proteins called heat shock proteins (HSPs). Although their expression is induced by several stimuli [1], they are commonly recognized as HSPs due to the first experiments showing their increased transcription after application of heat shock [2,3,4]. These proteins are molecular chaperones mainly involved in assisting protein transport and folding, assembling multimolecular complexes, and triggering protein degradation by proteasome. In addition, they play a crucial role in gene expression regulation, DNA replication, signal transduction, cell differentiation, apoptosis, cellular senescence or immortalization, and intercellular communications [5,6,7,8]. Heat shock proteins are classified according to their molecular weight in super heavy, 100, 90, 70, 60, 40, and small HSPs [9]. Although they are the most highly conserved, ubiquitous, and abundant proteins in all organisms, their cellular stress response can depend on the class and stimulus.

Physical exercise is considered a physiological stress factor able to stimulate a robust increase in different HSPs in several tissues, which appears to be fundamental in facilitating the cellular remodeling processes related to the phenomenon of adaptation [10].

Skeletal muscle, which represents about 40% of the body mass in humans, is a copious and plastic tissue [11,12,13,14], which largely responds to physical exercise and needs the HSP chaperoning system to control intracellular components and communications [15,16]. Indeed, it is widely accepted that exercise modulates the activity and the expression of HSPs in skeletal muscles depending on the characteristics of exercise program (i.e., type, intensity duration, and frequency) [17,18,19,20,21]. The exercise-induced changes in HSPs seem to have multiple cytoprotective effects on mitochondria, sarcoplasmic reticulum and cytoskeleton components [22,23,24], inhibitory effects on apoptosis [25], as well as a role in the maintenance of enzymatic activity, insulin sensitivity, and glucose transport [26,27].

Among the HSPs, special attention needs to be devoted to Hsp60 (HSPD1) and αB-crystallin (CRYAB, HSPB1), proteins constitutively expressed in the skeletal muscle, where they are known to be important in muscle physiopathology [28,29,30,31,32]. Therefore, given the emerging role in general health and the disease conditions of Hsp60 and CRYAB, here we provide a critical update on their role in skeletal muscle fibers after physical exercise, highlighting the control of their expression, their biological function, and their specific distribution within skeletal muscle fiber-types.

## 2. Fiber-Types of Skeletal Muscle and Its Adaptation to Physical Exercise

Skeletal muscle is a heterogeneous tissue containing fibers with different morphological, metabolic, and functional properties. Muscle fibers are typically identified by the expression of a multigene family of Myosin Heavy Chains (MHCs). Their sequence variability has been associated with their specific structure and function, but the codified actin-based motor proteins are conserved [33,34]. In the skeletal muscle they are classified as MHC-I, MHC-IIa, MHC-IIx, and MHC-IIb [33,34,35]. The MHC-I, IIa, and IIx fiber-types are expressed to variable degrees in both small animals (mice, rats, and rabbits) and human skeletal muscle. Instead, MHC-IIb fibers are solely expressed in small animals’ skeletal muscle [21]. In mammals, ‘‘hybrid’’ fibers (i.e., type I/IIa, IIa/x, IIx/b) can occur when different MHC transcripts coexist in a single fiber [12,36,37].

Although the structural and functional muscle requests define privileged associations between MHC isoforms, other common fiber-type classifications are used. One of them is related to the contractile response (or speed of contraction) distinguishing between slow or fast muscle fibers. In addition, cellular metabolism has been used as another parameter to distinguish the different types of myofibers with glycolytic metabolism and fibers with a massive mitochondrial presence and prevalent aerobic metabolism. Notably, MHC isoform expression correlates with fiber-type morphology, metabolism, and function [38,39]. In general terms, MHC-I-expressing fibers are small, rich in oxidative enzymes, slow in contraction, and have a greater resistance to fatigue, while MHC-IIb-expressing fibers are large, rich in glycolytic enzymes, and fast in contraction due to the developed sarcoplasmic reticulum that allows the rapid release of calcium ions, and a predominantly anaerobic metabolism [40,41]. Specifically, fiber contraction correlates with myosin ATPase activity with relative velocities of I < IIa < IIx < IIb [42,43]. Furthermore, skeletal muscle can be classified as postural or non-postural according to its function and the percentage of each fiber-type [44]. Although the four main types of skeletal muscle fibers are present in different percentages among the mammal species and within the same species therein, they are represented differently in the various structures of the musculoskeletal system (Table 1) [45,46,47,48]. Moreover, basal mitochondrial content has been shown to vary between species and is fiber-type-specific in mouse, rat, and human skeletal muscles [49].

One of the most surprising characteristics of the myofibers of the skeletal muscle is the high degree of plasticity as an adaptive response to physiological and non-physiological requests [50]. Plasticity is the ability of a tissue to modify its composition by adapting it to changing functional needs. Repeated, prolonged, or simple changes in functional requests can both quantitatively and qualitatively modify muscle tissue. These changes can affect the myofibrillary system, the sarcoplasmic reticulum, the proteins involved in regulating the concentration of intracellular calcium as well as the enzymatic systems involved in energy metabolism. The mechanism of adaptation of the functional requests is based primarily on the transformation of the MHC content, with a shift of the MHC fibers that affects the overall speed of contraction of the muscle. However, it is also possible that an increase in the speed of the same cellular type occurs in the absence of variations of the expressed MHC isoforms [51,52]. The increased contractile activity following physical exercise activates several signal pathways that lead to significant phenotypic changes such as MHC fiber transitions, enhanced mitochondrial biogenesis, and angiogenesis. Changes in the expression pattern of MHC isoforms and in the cross-sectional area (CSA) of the skeletal muscle cells, in response to different training protocols, are related to the changes in strength and power that the muscle undergoes [53]. For these reasons, physical exercise induces muscle hypertrophy that is followed by the upregulation of contractile elements’ synthesis. Literature data have shown the hypertrophic response for all three major types of fibers (MCH-I, IIa, and IIx) following resistance training, both in young subjects and in elderly subjects [54]. However, exercise-induced hypertrophy seems to affect fast muscle fibers more than type I fibers [55]. Exercise can therefore induce changes in the expression of MHCs, thus causing a switch from type IIb to IIx and IIa and, in rare cases, also to type I. In most cases, physiological adaptations to increased activity induce a switch to a more oxidative fast phenotype [35]. Allen et al. showed a significant increase in the percentage of fibers expressing MHC-IIa and a concomitant decrease in the percentage of fibers expressing MHC-IIb in mouse fast muscles after some weeks of wheel exercise [56]. The switch in the range IIb–IIx–IIa in mouse and rat muscle or in the range IIx–IIa in human muscle likely reflects the total amount of activity [57].

Human studies have shown that strength training induces an increase in type IIa and hybrid IIa/x fibers at the expense of fast IIx fibers. At the same time, there is an increase in the CSA in all types of myofibers, indicating a hypertrophic effect [58]. In fact, Kesidis et al. showed that type IIa muscle fibers in human skeletal muscle seem to have an enzymatic profile and a rate of contraction that makes these fibers more suitable for strength performance than fibers containing the MHC I isoform [59]. On the other hand, endurance training seems to induce fast-to-slow fiber transitions (from IIx to IIa, and in rare cases type I), while the CSA values remain unchanged. The physiological advantage is the greater transduction efficiency of the mechanical energy associated with the IIa fibers compared to the IIx fibers. However, the increased proportion of type I fibers could derive from different training protocols, causing a significant turnover and regeneration of fibers, and also including a regeneration of the peripheral nerve [60]. The absence of cellular damage can be explained by the lack of the expression of embryonic myosin isoforms and supports the results of the studies in which no variations in fiber-type I following training protocols were detected [61].

## 3. Stress Proteins: Heat Shock Protein 60

The HSPD1 gene comprises ~17 kb with 12 exons and it is localized at chromosome locus 2q33.1. This gene encodes a protein of 573aa corresponding to a molecular weight of 61.05 kDa known as HSP60 or Hsp60, also commonly referred to as Cpn60 [62]. Hsp60 belongs to group I of chaperonins [63]. Its ATP-dependent chaperon mechanism was thoroughly investigated for the prokaryotic homolog GroEL. Three structural domains were identified for GroEL: apical, intermediate, and equatorial. To carry out its chaperoning function, GroEL needs to generate a tetradecamer complex with its co-chaperon GroES (the homolog of Hsp10) [64]. Thus, the chaperon complex is made up of GroEL, structured in two rings with seven identical subunits, and GroES, which binds to the apical domains of GroEL to close the cage [65]. The chaperon mechanism is a multistep process that involves the unfolded protein binding to GroEL apical domains. Concomitantly, ATP binds to GroEL’s equatorial domain and its hydrolysis allows the conformational change (from trans to cis) of the GroEL apical and intermediate domains for the substrate encapsulation in the central cavity of the chaperon [66]. Consecutively, the dissociation of the cis-complex and the release of the folded protein, ADP, and GroES occurs [67]. Although the chaperon mechanism of the mammalian mitochondrial Hsp60-Hsp10 complex is similar, the solid-state structure appears as a symmetrical football-shaped complex by X-ray [68]. In humans, the mitochondrial Hsp60 exists as a homo-oligomer of seven subunits in equilibrium with very minor populations of monomers and double-ring tetradecamers [69,70]. In addition, it was demonstrated that Hsp60 single ring could perform chaperonin-mediated folding activity in vivo [71].

Hsp60 is constitutively expressed under philological conditions, so much so that its knockout is incompatible with life [72,73,74]. At the same time, Hsp60 expression is related to numerous etio-pathological conditions [75]. Hsp60 is mainly localized in the mitochondria. The mitochondrial import signal (MIS) at the N-terminus drives Hsp60 from the cytoplasm to the mitochondria [76]. Nevertheless, one third of Hsp60 is localized to the extra-mitochondrial sites, such as cytosol, plasma-cell membrane, inside exosomes, extracellular space, and circulation [77,78]. Inside the mitochondria, Hsp60 guarantees the correct folding of other mitochondrial proteins [7,79] as well as its “unfoldase” activity to stabilize misfolded and aggregated proteins, making provision for the mitochondrial biogenesis and protein homeostasis [80,81]. In addition, the mitochondrial Hsp60 directs the replication and transmission of mitochondrial DNA (mtDNA) [82,83]. Otherwise, the extramitochondrial Hsp60 is involved in intracellular protein trafficking [84] and peptide-hormone signaling [85]. Interestingly, the mitochondrial and the cytosolic Hsp60 have a contradictory role in pro-apoptotic and pro-survival mechanisms [86]. Whether Hsp60 is associated with carcinogenesis, specifically with tumor cell survival and proliferation, for certain tumors is used as a good diagnostic marker [87,88,89,90]. Therefore, Hsp60 exerts divergent roles in several physiological and pathological processes, and an understanding of its structural and functional biology aspires to draw novel pathways and to develop therapeutic strategies.

## 4. Stress Proteins: αB-Crystallin

HSPB1, or the CRYAB gene, encodes a 175-amino acid protein with a molecular mass of ~20 kDa [91]. CRYAB is a ubiquitous sHSP with highly conserved stretch that adopts a β-sandwich, immunoglobulin-like fold called the “α-crystallin domain (ACD),” which is a characteristic hallmark of the sHSPs family [92]. The ACD region is flanked by a less conserved and flexible N-terminal domain (NTD) and a C-Terminal extension (CTE), which are variable in length and sequence except for few conserved stretches [92,93].

Based on all findings related to CRYAB missense, truncating, and frame-shift mutations, specific functional roles of these domains have been hypothesized. Indeed, mutations within ACD domain (i.e., D109H/D109A, R120G) seem to interfere with the CRYAB oligomerization processes [94,95,96], while those within the CTE domain (i.e., 464delCT, R151X, G154S, L155fs_163X, R157H) seem to compromise CRYAB chaperone function [25,97,98,99,100,101]. Finally, mutations within the NTD domain seem to prevent the building of higher order oligomeric structures [102]. Thus, oligomeric assembly and chaperone activity of CRYAB is inter-dependent on its NTD, ACD and CTE domains.

As all sHSPs do, CRYAB shares in the property to form globular oligomer structures that in mammalian cells are characterized by molecular masses ranging from 50 to about 800 kDa. This ability, together with the well-known hetero-oligomerization property, is crucial factor in regulating the activity of this protein [103]. This hetero oligomer is probably unable to play an efficient protective role in stress conditions. However, the dissociation of the complex after-exposure to heat shock or oxidative stress suggests that it could bear new protein target recognition abilities and/or modulate those of the parental molecules [104].

Another intriguing property of sHSPs concerns its ability to be phosphorylated and therefore susceptible of control by several transduction pathways. Depending on the type and/or duration of various stimuli, the fraction of phosphorylated CRYAB ranges between 10% and 27% [105,106]. Different studies demonstrate that the phosphorylation of CRYAB shows a dual role that leads to either beneficial or deleterious outcomes depending on the extent and duration of stress and subsequent degree of phosphorylation; a phosphorylation at initial stage of stress is usually reversible and seems to provide a beneficial outcome, while prolonged stress can induce an irreversible phosphorylation which may lead to a deleterious outcome [107]. The CRYAB has three phosphorylation sites (S19, S45, and S59) at the NTD, which play a critical role in the protein functions. While the phosphorylation on S59 is mediated by p38 mitogen-activated protein kinase (p38 MAPK) and phosphorylation on S45 by the extracellular signal-regulated kinase 1/2 (ERK1/2) [106,108], the kinase responsible for phosphorylation on S19 is still unknown. Nevertheless, both unphosphorylated and phosphorylated forms of CRYAB are reported to be equally effective in preventing in vitro assembly of glial fibrillary acidic protein and vimentin in an ATP-independent manner [109]. In fact, during physiological or pathological stress both CRYAB content and phosphorylation can be modulated [29,110,111,112,113].

All aforementioned serine residues can be phosphorylated after various stimuli [106], but only a few studies have reported their contemporary involvement in muscle tissues [32,113,114,115,116]. To date, most of the available data are related to CRYAB expression and/or activation at Ser59 [29,110,117,118,119,120,121,122]. Moreover, the relationship between the phosphorylation of CRYAB and its chaperone activity is contradictory. Though in general the phosphorylation has an augmentative effect, it is possible that modulation of the activity upon phosphorylation might depend on the target protein and its interactions [123]. Further details about the phosphorylation of CRYAB in various physiological or pathological conditions can be found elsewhere [107,124].

In addition to being overexpressed in stress conditions, CRYAB shares the ability of having a tissue/cell-specific expression in the absence of stress, which can be detected in healthy adults as well as during organism development [125]. In mammalian cells, CRYAB is constitutively expressed in tissue with high rates of oxidative metabolism, including the cardiac and skeletal muscle [126]. The significance of the constitutive expression of this sHSP is probably linked to the protection of the cells against chronic stress or to a specific function in a particular tissue.

## 5. Hsp60 in Skeletal Muscle Fibers

The chaperoning systems that participate in controlling cellular homeostasis have been detected in skeletal muscle. Small Hsp, Hsp60, Hsp70, and Hsp90 play a significant role in muscle adaptation [17,127]. Nevertheless, Hsp60 was not deeply investigated after physical exercise, which, as we have already discussed, influences muscle homeostasis [77]. Hsp60 and exercise correlation appears to be rational, but literature data are restricted and sometimes controversial (Table 2). Morton et al. [128] demonstrated that aerobically trained men had significantly higher resting levels of Hsp60 in the vastus lateralis muscle (high percentage of fibers I and IIa) compared to untrained subjects, suggesting Hsp60 as a molecular marker of physiological adaptation to aerobic exercise. However, it has been demonstrated that in the human vastus lateralis muscle the highest mitochondrial content is showed by type I fibers followed by type IIa > IIx (Table 1) [49]. At the same time, an acute single bout of endurance training that is considered an aerobic exercise did not increase the basal Hsp60 protein levels in the same muscle [128]. Thus, chronic training has the capacity to increase Hsp60 expression, whereas a single bout of exercise does not. Hsp60 expression is stimulated by endurance, resistance, and mixed training, but its fiber specificity is still debated. Hsp60 expression in the vastus lateralis of healthy active people with different training backgrounds was considered not to be fiber-type specific [129]. In agreement, Ogata et al. [130] and Soares Moura et al. [131] did not show significant differences in Hsp60 levels in the plantaris and gastrocnemius, both rich fiber IIx muscles, of male rats after endurance training. On the other hand, several groups, including ours, demonstrated fiber-type specificity after training in specific muscles. Mattson et al. [132] showed that female rats trained with an endurance protocol for 8 weeks displayed significantly higher levels of Hsp60 in the muscle plantaris, which is rich in fiber-type IIb [46]. No difference of Hsp60 levels was detected in the rich fiber I muscle soleus in endurance-trained rats compared to the untrained group [47,132]. Hsp60 fiber-type I specificity was reported by Samelman [133], who showed increased basal levels of Hsp60 in the soleus and not in the lateral gastrocnemius of endurance trained rats. In agreement, our group noted Hsp60 fiber-specific expression in healthy male BALB/c mice trained for 45 days on the treadmill. Specifically, higher levels of Hsp60 were observed in type I and IIa muscle fibers, while type IIx and IIb fibers showed a constitutive expression of this chaperonin [134]. Therefore, increased levels of Hsp60 were reported after six weeks of endurance training, mainly in red gastrocnemius and soleus muscles, which are particularly rich in type I and IIa fibers [134]. We also correlated this physiological adaptation to an increased expression of peroxisome proliferator-activated receptor gamma coactivator 1 alpha (PGC-1α), which triggers the mitochondrial biogenesis and thereby avoiding the cytotoxic effects [134,135,136]. Finally, increased levels of Hsp60 were observed in the soleus muscle of male mice and the Extensor Digitorum Longus (EDL) muscle of female mice after an acute single bout of endurance training [48].

## 6. αB-Crystallin in Skeletal Muscle Fibers

CRYAB is a protein highly expressed in slow and fast fibers of adult skeletal muscle where it is associated with actin microfilaments at level of Z-bands [98]. Different lines of evidence suggest that this sHSP protects the skeletal muscle from heat, oxidative, and mechanical stresses produced during physical activity [32,110,137,138,139].

At the molecular level, it has been demonstrated that the alteration of any of the three major components (i.e., microfilaments, microtubules, and intermediate filaments) results in a specific activation of p38MAPK and MAPKAP kinases 2 and 3 and the phosphorylation of CRYAB [115]. Our group also demonstrated that a reversible redox unbalance, which represents one of the main stimuli under different circumstances, including physical activity, induces CRYAB expression through a JNK-mediated transcriptional mechanism in myogenic cells [110]. On the one hand, the increased level of CRYAB and its phosphorylation determine its translocation to the myofilaments where it binds titin, desmin, vimentin, nebulette, filamin A, and the inactive precursor of caspase 3, leading to the stabilization of the myofilament and to the inhibition of apoptosis [29,32]; on the other side, it enhances NFκB activity, which translocates into the nucleus, inducing the expression of genes involved in various biological events such as growth, differentiation, and cell death [29,140,141].

Moreover, CRYAB appears to have a role in regulation of apoptosis during heat shock, oxidative stress, and ischemia. This protein is able to prevent apoptosis by several mechanisms such as the inhibition of RAS-initiated RAF/MEK/ERK signaling pathway [142], or downstream, blocking the BAX and Bcl-2 translocation from cytoplasm to mitochondria [143], as well as interacting with p53 to retain it in the cytoplasm [144] or inhibiting autocatalytic maturation of caspase-3 [145].

A recent finding has established that CRYAB is necessary for a correct skeletal muscle homeostasis via modulation of Argonaute 2 (Ago2) activity [120], a protein with endonuclease activity. It belongs to the central core of RNA-induced silencing complex (RISC), capable to repress the translation of mRNA into protein via a variety of mechanisms such as removal of the of the 5-7-methylguanylate cap (m7G), deadenylation of the 3′-poly(A) tail, and miRNA site-directed endonuclease cleavage of the mRNA [146,147]. This result indicates that CRYAB functions as a positive allosteric regulator of Ago2/RISC. In fact, the absence of CRYAB results in the unbalance of hypertrophy-atrophy axis toward atrophy with an excessive miRNA loading into Ago2/RISC [120].

The study of the distribution of CRYAB in skeletal muscle dates back to the early 1990s [148,149]. Subsequently, other authors investigated the distribution of this sHSP in skeletal muscle even in pathological conditions [150,151]. In mammalian studies, all studies agree that CRYAB is predominantly expressed in oxidative muscle fibers (type I and IIa) [148,149,150,151].

Despite the paucity of the number of studies in the field of physical activity and distribution of CRYAB in muscle fibers, exercise-based studies analyzing the modulation of this sHSP in humans and rodents reported a modulation of CRYAB dependent from the characteristics of exercise. It was observed that this sHSP was up-regulated depending on the damaging nature of exercise. Indeed, in non-damaging conditions, the CRYAB protein level was unchanged independently from the skeletal muscle examined [21,32,129,152], whereas following an exercise associated with damage to strain-bearing cell structures, CRYAB was significantly up-regulated [129,153,154,155] (Table 3). In contrast, the phosphorylation form of CRYAB was found increased in muscles with the highest percentage of type I fibers, independently from the type (endurance or resistance) and nature (damaging or not) of exercise [32,129,155,156] (Table 3).

The analysis of fiber-type-specific expression/activation of CRYAB has revealed that its redistribution within skeletal muscles is influenced by different physical activity regimes [32,129,156]. In particular, a higher staining intensity of CRYAB was found in type I compared to type II fibers, which was in line with previous results obtained in rodents, where a stronger CRYAB expression was found in slow oxidative muscle fibers [157,158]. However, Folkesson and colleagues show a similar staining intensity of CRYAB between type II and type I fibers in well-trained endurance athletes [129]. A similar fiber type-specific increase in CRYAB has been shown in rabbit tibialis anterior muscles following 21 days of continuous low-frequency motor nerve stimulation, which keeps the fiber-specific pattern of CRYAB from being expressed only in type I fibers and in a subpopulation of type II fibers, to be expressed in nearly all fibers [157]. Recently, our group has shown that following an acute non-damaging endurance exercise, the phosphorylation level of this sHSP was significantly increased only in skeletal muscle with a higher amount of type I and IIA/X myofibers. In particular, the phosphorylation level of CRYAB was apparently related to the percentage of type I and IIA/X myofibers contained in slow twitch and mixed muscles (i.e., the soleus and red gastrocnemius). In support of the aforementioned data, Jacko and colleagues demonstrated that a multiple set of resistance exercises increases the phosphorylation level of CRYAB in type I myofibers of the vastus lateralis independently from the load volume, while in the type II fibers the phosphorylation was observed only after high force demanding loadings [156].

These results could reflect the role of CRYAB in counteracting homeostatic perturbations, including mechanical, thermal, and oxidative stress induced by physical exercise, as well as to reinforce the idea that the phosphorylation of CRYAB in these tissues possibly reflects the level of stress experienced by the muscle.

## 7. Summary and Conclusions

There is substantial evidence that regular physical activity promotes health and healthy aging [118,138,159,160,161]. In particular, we know from our work and that of other laboratories that trained subjects show an adaptive response at both systemic and cellular levels through the modulation of antioxidants and stress-induced proteins, such as HSPs, as well as an improvement of other parameters strictly correlated to the healthy status [162,163,164,165,166,167,168]. Moreover, it is known that exercise induced-adaptation in mammalian skeletal muscle, and in other tissues, is related to training mode and can be fiber-type specific.

In this review, we highlighted for the first time the regulation and function of Hsp60 and CRYAB, as well as their different expression/activation in skeletal muscle fibers in response to exercise. The determination of a fiber-type-specific expression of Hsp60 and CRYAB could be important to comprehend the impact of different volumes of training regimes on myofiber types and explains the complex picture of exercise-induced mechanical strain and damaging conditions on fiber population.

In summary, Hsp60 was found to be exclusively up-regulated in skeletal muscles rich in slow and fast oxidative fibers that were aerobically trained (endurance exercise), with a distribution not fiber-specific following other training regimes (e.g., resistance training). Similarly, CRYAB was mainly present in type I fibers, with an increase in its concentration in type II fibers depending on the individual’s fitness level. The phosphorylated form of this protein was more present in skeletal muscle with a higher amount of type I and IIA/X myofibers following endurance exercise, while its fiber-specificity appears to be dependent on the workload during resistance exercises.

This parallel behavior between Hsp60 and CRYAB within fiber-type distribution could be due to the involvement of CRYAB in the response to oxidative stress stimuli and the role of Hsp60 during the mitochondrial metabolism. In fact, the direct correlation between CRYAB and oxidative enzymes was already evident in the first studies on its cellular distribution [169].

Although further studies are recommended to clarify the role of these HSPs in muscle physiology, we strongly believe that this knowledge could lead to a better personalization of training protocols with an optimal non-harmful workload in people with different needs and healthy status. In the long term, the new knowledge will be expanded so to accumulate specific insight towards the effectiveness of muscle contraction patterns and the adaptive state of frequently loaded skeletal muscle, to be utilized in prevention, rehabilitation, and elite sports.

## Figures and Tables

**Table 1 biology-10-00077-t001:** Fiber-type distribution and mitochondrial content in different skeletal muscles.

Skeletal Muscles	Species	Gender	Main Fiber-Types %	Mitochondrial Content	References
Vastus Lateralis	Human	Male	I 49–IIa 42%	I > IIa > IIx	[44,48]
Plantaris	Rat	Male	IIx 45–IIa 21%	IIa > I > IIx > IIb	[44,48]
Plantaris	Rat	Female	IIb 46–IIx 40%		[45]
Soleus	Rat	Male	I 97%	IIa > I > IIx > IIb	[44,48]
Soleus	Rat	Female	I 99%		[46]
Gastrocnemius	Rat	Male	IIx 43–IIb 26%		[44]
Soleus	Mouse	Male	IIa 49–I 31%		[44]
Soleus	Mouse	Female	I 49–IIa 35%		[47]
EDL	Mouse	Male	IIb 63–IIx 18%		[47]
EDL	Mouse	Female	IIx 37–IIb 35%		[47]
Gastrocnemius	Mouse	Male	IIb 56–IIa 21%	IIa > IIx > I > IIb	[44,48]

EDL, extensor digitorum longus; I, type I; IIa, type IIa; IIx, type IIx; IIb, type IIb.

**Table 2 biology-10-00077-t002:** Hsp60 expression levels in different skeletal muscles after physical exercise.

Species/Strain	Gender (Age)	Skeletal Muscles	Protocol Training	Hsp60 Levels	References
Human	Male (28 ± 6 yrs)	Vastus Lateralis	Endurance	↑	[128]
Human	Male (28 ± 6 yrs)	Vastus Lateralis	Acute exercise	=	[128]
Rat/Wistar	Male (4 months)	Plantaris	Endurance	=	[130]
Rat/Wistar	Female (ns)	Plantaris	Endurance	↑	[133]
Rat/Fischer 344	Male (10 months)	Soleus	Endurance	↑	[133]
Rat/Wistar	Female (ns)	Soleus	Endurance	=	[132]
Rat/Wistar	Male (ns)	Gastrocnemius	Endurance	=	[131]
Rat/Fischer 344	Male (10 months)	Gastrocnemius	Endurance	=	[133]
Mouse/BALB/c	Male (7 weeks)	Soleus	Endurance	↑	[134]
Mouse/BALB/c	Male (12 weeks)	Soleus	Acute exercise	↑	[48]
Mouse/BALB/c	Female (12 weeks)	Soleus	Acute exercise	=	[48]
Mouse/BALB/c	Male (12 weeks)	EDL	Acute exercise	=	[48]
Mouse/BALB/c	Female (12 weeks)	EDL	Acute exercise	↑	[48]
Mouse/BALB/c	Male (7 weeks)	Gastrocnemius	Endurance	=	[134]

Hsp60, heat shock protein 60; EDL, extensor digitorum longus; arrow, increased levels of Hsp60; =, no difference; ns, not specified; yrs, years.

**Table 3 biology-10-00077-t003:** CRYAB expression levels in different skeletal muscles after physical exercise.

Species/Strain	Gender (Age)	Skeletal Muscles	Protocol Training	CRYAB Levels (Protein)	CRYAB (S59) Levels	References
Human	Male (25 ± 6 years)	Vastus Lateralis	Resistance	↑	NA	[129]
			Endurance	=	NA	
Human	Male/Female (21–37 years)	Biceps brachii	Endurance	↑	NA	[153]
Human	Male (20.3 ± 0.8 years)	Vastus Lateralis	Endurance	↑	NA	[154]
Human	Male (24 ± 4 years)	Vastus Lateralis	Endurance	=	NA	[21]
Human	Male (24 ± 3 years)	Vastus Lateralis	Endurance	=	↑	[152]
Human	Male (24.8 ± 3.8 years)	Vastus Lateralis	Resistance	=	↑	[156]
Mouse/BALB/c	Male (7 weeks)	Soleus Red Gastrocnemius	Endurance	=	↑	[32]
Mouse/C57BL/6	Male (3 months)	EDL	Resistance	=	↑	[155]

CRYAB, αB-crystallin; CRYAB(S59), phospho-αBcrystallin (S59); EDL, extensor digitorum longus; arrow, increased levels of CRYAB/CRYAB(S59); =, no difference; NA, not applicable; yrs, years.

## Data Availability

Not applicable.

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
