# Peer review of "Function and Fiber-Type Specific Distribution of Hsp60 and αB-Crystallin in Skeletal Muscles: Role of Physical Exercise"

_biology, 2021, doi:10.3390/biology10020077_

Round 1

Reviewer 1 Report

The review by Amico et al. with the title 'Function and fiber-type specific distribution of Hsp60 and aB-crystallin in skeletal muscles: rope of physical exercise' is well written and informative and would be suitable for publication in the journal Biology. It summarises the current state of research regarding the role of exercise in muscle fiber plasticity and how heat shock proteins are involved in this process. The authors make a good argument for their view about the importance of HSPs in muscle development through training and outline how future research and a better application of the existing knowledge can help to design more efficient, personalised training protocols for patients and sports persons.  

below a few minor comments and corrections.

correction in line

39/40: ...stress condition by upregulating the expression and/or activation of ...

49: HSPs

52: stimulate

352: proper citation for Folkesson et al.

363: proper citation for Jacko et al.

comments to line

76-86: it would be good to add some reference to the statements regarding the muscle phenotypes found in vitro and in vivo and regarding the regulation of myofibers development.

Author Response

Dear Editor of Biology

It is a pleasure for me to send you the revised file of our review entitled “Function and fiber-type specific distribution of Hsp60 and αB-crystallin in skeletal muscles: role of physical exercise”.

We thank you very much for give us the opportunity to improve our work.

Here is a detailed answer to Reviewer 1.

Authors point-by-point reply to Reviewers’ Comments

Reviewer 1 comment  #1 (R1C#1). Below a few minor comments and corrections.

correction in line

39/40: ...stress condition by upregulating the expression and/or activation of ...

49: HSPs

52: stimulate

352: proper citation for Folkesson et al.

363: proper citation for Jacko et al.

Authors’ Reply (AR): We thanks the reviewer for these comments because he/she gives us the opportunity to correct the mistakes. We modified the mistakes.

(R1C#2). Comments to line

76-86: it would be good to add some reference to the statements regarding the muscle phenotypes found in vitro and in vivo and regarding the regulation of myofibers development.

  1. AR. We thanks the reviewer for this comment. We revised the sentence.

All changes were left tracked in the revised text (highlighted in red). We hope that this new version of our review accomplish your request. We want to thank you once again and we hope the review is now acceptable for publication.

Best regards.

Dr Rosario Barone

Department of Biomedicine, Neuroscience and Advanced Diagnostics (BiND), Human Anatomy

Section, University of Palermo, via del Vespro 129, 90127, Palermo, Italy;

email: rusbarone@hotmail.it

Reviewer 2 Report

Thank you for the invitation to review “Function and fiber-type specific distribution of Hsp60 and αB-crystallin in skeletal muscles: role of physical exercise” by D’Amico et al. This is an interesting and topic review article, highlighting the role of HSPs in skeletal muscle in the context of exercise training.

The review clearly focuses on the role of HSPs in an intracellular context, with a very brief hint to extracellular localisation. I feel for the sake of completeness it would improve the manuscript if the authors could provide some detail and insight on extracellular HSPs and inference to exercise (adaptation).

Line 43: The discovery of HSPs should be attributed to Ferrucio Ritossa, with his seminal paper in 1962.

Line 43: Typo, it should read …molecular chaperones…

Line 72: The opening sentence is overly short and not grammatically correct, revise.

Lin 72 – 86: This whole paragraph is lacking and references. Further, the paragraph is poorly written and feels quite vague – please revise accordingly.

It would be nice for the authors to perhaps give more of an insight on the future perspectives of HSPs in the context of exercise training – where is the field going?

Line 386: Should begin “In summary…”

Table 2: Where possible, provide detail on human demographics and strain/age of rats and mice.

Table 3: It is possible add further demographic details? Age would be particularly useful. Further, clarify the increase in levels of the CRYAB, are they protein or gene expression measures?

Author Response

Dear Editor of Biology

It is a pleasure for me to send you the revised file of our review entitled “Function and fiber-type specific distribution of Hsp60 and αB-crystallin in skeletal muscles: role of physical exercise”.

We thank you very much for give us the opportunity to improve our work.

Here is a detailed answer to Reviewer 2.

Reviewer 2 comment  #1 (R2C#1). The review clearly focuses on the role of HSPs in an intracellular context, with a very brief hint to extracellular localisation. I feel for the sake of completeness it would improve the manuscript if the authors could provide some detail and insight on extracellular HSPs and inference to exercise (adaptation).

  1. We thank the reviewer for this comment. To date, there are limited data on extracellular HSPs and inference to exercise (adaptation), we have already covered this aspect in two other reviews (Trovato E, Di Felice V, Barone R. Extracellular Vesicles: Delivery Vehicles of Myokines. Front Physiol. 2019 May 7; 10:522. Marino Gammazza A, Macaluso F, Di Felice V, Cappello F, Barone R. Hsp60 in Skeletal Muscle Fiber Biogenesis and Homeostasis: From Physical Exercise to Skeletal Muscle Pathology. Cells. 2018 Nov 22;7), so we preferred not to deepen and repeat this aspect.

(R2C#2). Line 43: The discovery of HSPs should be attributed to Ferrucio Ritossa, with his seminal paper in 1962.

  1. We thank the reviewer for this comment. We added the reference.

(R2C#3). Line 43: Typo, it should read …molecular chaperones…

  1. We modified the mistake.

(R2C#4). Line 72: The opening sentence is overly short and not grammatically correct, revise.

  1. We thank the reviewer for this comment. We revised the sentence.

(R2C#5). Lin 72 – 86: This whole paragraph is lacking and references. Further, the paragraph is poorly written and feels quite vague – please revise accordingly.

  1. We thank the reviewer for this comment. We revised the sentence.

(R2C#6). It would be nice for the authors to perhaps give more of an insight on the future perspectives of HSPs in the context of exercise training – where is the field going?

  1. As already explained in the paragraph of the last section (Summary and Conclusions), we strongly believe that our review highlight an other important aspect of HSP60 and CRYAB related to muscle physiology, which could lead to a better personalization of training protocols with an optimal non-harmful workload in people with different needs and healthy status. Moreover, in the next future this new knowledge will be expanded so to accumulate specific insight towards the effectiveness of muscle contraction patterns and the adaptive state of frequently loaded skeletal muscle, to be utilized in prevention, rehabilitation, and elite sports.

(R2C#6). Line 386: Should begin “In summary…”

  1. We revised the sentence.

(R2C#7). Table 2: Where possible, provide detail on human demographics and strain/age of rats and mice.

  1. The table 2 has been integrated with detail on human demographics and strain/age of rats and mice.

(R2C#8). Table 3: It is possible add further demographic details? Age would be particularly useful. Further, clarify the increase in levels of the CRYAB, are they protein or gene expression measures?

  1. As suggested by the Reviewer, we included the demographic details and specify in the column that all results reported on CRYAB level were referred to its protein form (see Table 3).

All changes were left tracked in the revised text (highlighted in red). We hope that this new version of our review accomplish your request. We want to thank you once again and we hope the review is now acceptable for publication.

Best regards.

Dr Rosario Barone

Department of Biomedicine, Neuroscience and Advanced Diagnostics (BiND), Human Anatomy

Section, University of Palermo, via del Vespro 129, 90127, Palermo, Italy;

email: rusbarone@hotmail.it
